# The Effect of HIV and Antiretroviral Therapy on Drug-Resistant Tuberculosis Treatment Outcomes in Eastern Cape, South Africa: A Cohort Study

**DOI:** 10.3390/v15112242

**Published:** 2023-11-10

**Authors:** Brittney van de Water, Nadia Abuelezam, Jenny Hotchkiss, Mandla Botha, Limpho Ramangeola

**Affiliations:** 1Connell School of Nursing, Boston College, Chestnut Hill, MA 02467, USA; nadia.abuelezam@bc.edu; 2Morrissey College of Arts and Sciences, Boston College, Chestnut Hill, MA 02467, USA; hotchkij@bc.edu; 3Eastern Cape Department of Health, Marjorie Parish Tuberculosis Hospital, Port Alfred 6170, South Africa; mbotha@gmail.com; 4Eastern Cape Department of Health, Jose Pearson Drug Resistant Tuberculosis Hospital, Port Elizabeth 6055, South Africa; limphojr@yahoo.com

**Keywords:** HIV and TB co-infections, Sub-Saharan Africa, rifampicin-resistant TB, epidemiology, antiretroviral therapy

## Abstract

South Africa has a dual high burden of HIV and drug-resistant TB (DR-TB). We sought to understand the association of HIV and antiretroviral therapy status with TB treatment outcomes. This was a retrospective chart review of 246 patients who began treatment at two DR-TB hospitals in Eastern Cape, South Africa between 2017 and 2020. A categorical outcome with three levels was considered: unfavorable, transferred out, and successful. Descriptive statistics and logistic regression were used to compare the individuals without HIV, with HIV and on antiretroviral therapy (ART), and with HIV but not on ART. Sixty-four percent of patients were co-infected with HIV, with eighty-seven percent of these individuals on ART at treatment initiation. The majority (59%) of patients had a successful treatment outcome. Twenty-one percent of patients transferred out, and an additional twenty-one percent did not have a successful outcome. Individuals without HIV had more than three and a half times the odds of success compared to individuals with HIV on ART and more than ten times the odds of a successful outcome compared to individuals with HIV not on ART (OR 3.64, 95% CI 1.11, 11.95; OR 10.24, 95% CI 2.79, 37.61). HIV co-infection, especially when untreated, significantly decreased the odds of treatment success compared to individuals without HIV co-infection.

## 1. Introduction

Tuberculosis (TB) is the leading cause of infectious-disease-related death worldwide, and drug-resistant TB (DR-TB) poses a dual threat to public health and to efforts to overcome the growing challenge of antimicrobial resistance [1,2]. The burden of DR-TB in Africa is poorly measured, with only 51% of countries formally collecting data [3]. Using models based on World Health Organization (WHO) data, it is estimated that 42% of DR-TB cases occur in just two countries: Nigeria and South Africa [3]. South Africa also bears a dual burden of HIV, where up to 73% of all individuals with TB are co-infected with HIV [4]. The rate for non-co-infected TB deaths in South Africa is 38.5 per 1000 people, while HIV-associated TB deaths can be as high as 121.7 per 1000 people for co-infected patients, with DR-TB patients experiencing even higher mortality rates [4].

Providing care for individuals with DR-TB is complex, especially for those co-infected with HIV. Ensuring services are available is paramount, and providing a proper diagnosis and prompt treatment initiation for both TB and HIV is critical [5,6]. To explore the relationship between HIV, ART use, and DR-TB outcomes, we aimed to assess the effect of HIV infection and ART status on DR-TB treatment outcomes in two DR-TB hospitals in Eastern Cape, South Africa.

## 2. Materials and Methods

### 2.1. Demographics

This retrospective cohort included patients of any age with a documented treatment outcome for DR-TB at two DR-TB hospitals in Sarah Baartman District, Eastern Cape, South Africa, who initiated treatment between January 2015 and August 2018. There were 21 individuals who were excluded due to their lack of documented treatment outcome, 7 were excluded for their lack of documented HIV status, and 9 were excluded for their lack of documented ART status, leading to a final sample size of 246 (Figure 1).

### 2.2. Setting

Sarah Baartman District, Eastern Cape, South Africa has one of the highest rates of TB, DR-TB, and TB/HIV in the country and globally [7]. Its two DR-TB hospitals serve a catchment area of over 479,923 people in the Sarah Baartman District and 1,263,091 people in the Nelson Mandela Metro area [8,9]. As of 2011, the median annual household income in Eastern Cape was ZAR 64,539 (USD 3364.56), with 52.6% of adults reporting employment, and 22.5% reporting completing secondary education as of 2021 [10,11].

### 2.3. Data Sources and Study Measures

Medical charts were obtained with permission from hospital medical directors and support from hospital staff to extract patients’ information, including age, sex, HIV status, antiretroviral (ART) status if HIV infected, type of TB, previous TB drug history, if this was a patient’s first episode of TB, classification of disease (pulmonary versus extrapulmonary), comorbidities (including hypertension, diabetes, and hearing loss), body mass index, pregnancy status, and information about their TB diagnosis, such as if GeneXpert, TB microscopy, line probe assays (LPAs), TB culture, or drug sensitivity tests (DSTs) were completed. We also collected, among individuals who were HIV-positive, whether they were taking prophylactic co-trimoxazole. For social histories, we included the hospital in which individuals were treated (suburban or urban), patients’ education level, income, smoking status, alcohol use, and if they had any household contacts at the time of diagnosis. All data were entered into and downloaded from REDCap, an electronic data management system [12].

Treatment outcome definitions are in accordance with the WHO 2013 revised definitions and reporting framework for tuberculosis [13]. Composite outcomes were created: success, non-success, and transferred out. Success was defined as a patient being cured or completing treatment. Non-success was defined as treatment failure, death, and loss to follow-up. Transferred out was the outcome given to patients when clinicians provided documentation for the patient to transfer to another health facility. Given these patients did not have a definitive ‘final’ treatment outcome but rather transferred care on a positive trajectory with intentions of continuity, it was decided to keep this outcome separate.

### 2.4. Statistical Analysis

Descriptive statistics were used to report the association between HIV and ART statuses (HIV infected, not on treatment; HIV infected, on treatment; and HIV not infected) and the measured demographic and clinical covariates, using ANOVA and Chi-Square tests. Demographic variables included age, sex, education, income, and hospital location. Clinical variables included TB type, type of DR-TB, past TB history, classification of disease, history of hypertension, diabetes, hearing loss, prophylactic cotrimoxazole use, body mass index, pregnancy status, GeneXpert, LPA, DST, TB culture, hospital attended, education history, income, smoking status, alcohol use, and if there were any household contacts. Descriptive statistics (frequency tables) were also used to report treatment outcomes according to HIV and ART statuses.

Univariate logistic regression models were used to estimate the odds ratios for treatment outcomes, i.e., (1) transfer out versus non-success and (2) success versus non-success. Predictors of interest assessed individually included age, sex, HIV status, hospital attended, number of years of education, body mass index, smoking status, alcohol use, and income. These are all known predictors of interest influencing DR-TB treatment outcomes [14,15,16,17,18]. We decided to dichotomize age to increase the power due to the small number of children <16 years in the dataset. All analyses were performed using SAS version 9.4.

## 3. Results

### 3.1. Demographics

Among all 246 patients, the mean age was 36 years (SD 13.4 years) and 55% of patients were male (*p* = 0.12) (Table 1). One hundred and fifty-seven (64%) of the patients were co-infected with HIV. The majority of patients had either rifampicin-resistant TB (39%), multidrug-resistant TB (29%), or extremely drug-resistant TB (22%) (*p* = 0.13). Nearly half (47%) of the patients had no previous TB history (*p* = 0.11). At least 81% of the patients had pulmonary TB (*p* = 0.20). A total of 13% of the individuals reported hypertension (*p* = 0.09), 7% reported diabetes (*p* = 0.01), and 7% reported hearing loss (*p* = 0.49). The majority of individuals with an HIV infection were taking ART (87%). Eighty percent of individuals with HIV on ART were also taking cotrimoxazole, while fifty percent of patients with untreated HIV were taking prophylactic cotrimoxazole (*p* < 0.01). Two (1.8%) individuals were pregnant (*p* = 0.26). Less than half of the individuals (48%) had a GeneXpert test completed (*p* = 0.04). TB microscopy was performed for 69% of the individuals (*p* < 0.01), and first-line LPA was completed in 39% of the patients (*p* = 0.09), while second-line LPA was completed in 36% of the patients (*p* = 0.15). TB cultures were completed in 61% of the patients (*p* < 0.01). Twenty-two percent of patients had a first-line DST (*p* = 0.04) and twenty-three percent had a second-line DST (*p* = 0.05).

Fifty-seven percent of patients were seen at one of the two hospitals, which was the more urban hospital, closer to a large metro area. At this hospital, more patients with HIV were started on ART and there was a larger proportion of patients who were HIV-negative (*p* < 0.01). There were no differences among patients regarding education or income; the majority (85%) of patients had formal schooling (mean 8.9 years, STD 3.1), and 55% of patients reported no source of income (*p* = 0.58). Participants reported a mean of 1.73 (STD 1.62) dependents, and 96% reported household contacts, with significantly fewer individuals with untreated HIV reporting no household contacts (*p* = 0.03). Most patients did not smoke (60%), with significantly fewer individuals with HIV smoking compared to those who were HIV-negative (*p* = 0.01). Over half (52%) of the individuals reported not drinking alcohol (*p* = 0.39).

### 3.2. Treatment Outcomes

Most individuals (59%), regardless of their HIV and ART statuses, had a successful DR-TB treatment outcome (Table 2). However, the success rates steadily declined based on their HIV and ART statuses; individuals without HIV had a success rate of 72%, those with treated HIV had a success rate of 55%, and those with untreated HIV had a success rate of 25%. Of the 144 individuals with a successful outcome, 118 were bacteriologically cured while 26 completed treatment. Among the 21% of individuals with an unsuccessful DR-TB treatment outcome, the non-success rates steadily increased based on HIV and ART statuses; those without HIV had a non-success rate of 11%, those with treated HIV had a non-success rate of 24%, and those with untreated HIV had a non-success rate of 40%. Of the fifty-one individuals with an unsuccessful DR-TB treatment outcome, one failed treatment, thirty-nine died, and eleven were lost to follow-up. A total of 51 individuals (21%) transferred out of their hospital facility while still on treatment. Among the twenty-one individuals not included in the regression analysis, ten were still on treatment, sixteen had a missing outcome, and three moved out.

### 3.3. Regression Analysis

Comparing the estimated odds ratios for the individuals with a successful DR-TB treatment outcome versus those with an unsuccessful DR-TB treatment outcome, HIV status and hospital were significant predictors (Table 3). The odds of success were 10.24 times higher for individuals without HIV compared to individuals with untreated HIV (OR 10.24, 95% CI 2.79, 37.61). Additionally, the odds of success were 3.64 times higher for individuals with treated HIV compared to individuals with untreated HIV (OR 3.64, 95% CI 1.11, 11.95). Finally, the odds of success were nearly seven times higher at the urban hospital compared to the suburban hospital (OR 6.97, 95% CI 3.46, 14.04).

Additionally, when comparing the odds ratios for the individuals with a successful DR-TB treatment outcome versus those with death, HIV status was an even more significant predictor (Table 4). The odds of success were 13.06 times higher for individuals without HIV compared to individuals with untreated HIV (OR 13.06, 95% CI 3.21, 53.11). Additionally, the odds of success were 4.10 times higher for those with treated HIV compared to individuals with untreated HIV (OR 4.10, 95% CI 1.22, 13.72).

Comparing the estimated odds ratios for individuals who transferred out versus those experiencing unsuccessful DR-TB treatment outcomes, the odds of transfer out (versus non-success) were 2.56 times higher for individuals at the suburban hospital compared to the urban hospital (OR 0.39, 95% CI 0.16, 0.99) (Table 5). No other predictors were significant for transfer out versus non-success.

## 4. Discussion

Patients without HIV had 10 times higher odds of successful DR-TB treatment outcomes compared to those with untreated HIV. Even among patients with HIV, those who were on ART still had three and a half times higher odds of successful DR-TB treatment outcomes compared to those with untreated HIV. These results are similar to a 2022 study among patients in the Eastern Cape with drug-sensitive TB, where individuals without HIV had nearly five times greater odds of successful treatment outcomes compared to individuals with untreated HIV. However, our odds of success (versus non-success) were nearly double among this cohort with DR-TB than with DS-TB (OR 10.24 versus 4.98) [20]. In a systematic review of treatment outcomes and antiretroviral uptake in DR-TB patients, the cure rate ranged from 26 to 68%, the death rate ranged from 18 to 34%, and the default rate (or loss to follow-up) ranged from 1 to 22% [21]. Similar to our study, unsuccessful treatment was typically higher among individuals with HIV co-infection, with the ratio of treatment success to non-success being approximately 2:1 among HIV-positive patients and 3:1 among HIV-negative patients. However, in contrast to our study, this systematic review found that uptake of ART did not affect the TB cure rate among co-infected patients—cure outcomes ranged from 28 to 54% among patients on ART and from 22 to 58% among those not on ART [21]. Neither study accounted for the timing of ART initiation in relation to DR-TB treatment initiation, warranting further investigation.

In addition to ART status, hospitals were also a significant factor in treatment success in this study. The odds of success were nearly seven times higher at the urban hospital compared to the suburban hospital. This could have been due to a multitude of factors. All individuals with HIV at the urban hospital were initiated on ART (i.e., 0/77 had untreated HIV at the urban hospital compared to 20/80 individuals infected with HIV at the suburban hospital). This could be due to patient hesitancy or the timing of when a patient discovered their HIV status; however, the urban hospital had a robust program in place to initiate all HIV-infected individuals on ART (as per the hospital director). Integration of care, such as DR-TB and HIV services, is necessary and highly sought after in settings with a high prevalence of both diseases. However, the implementation of such service integration, and delivery of evidence-based practices in many high-burden settings, is often elusive [22,23,24]. Also, significantly more patients without HIV infection were seen at the urban hospital (72% versus 28%). However, a patient had 2.5 times greater odds of being transferred out (i.e., down-referral to a decentralized setting or a clinic closer to a patient’s home) at the suburban hospital than at the urban hospital. This indicates a higher prioritization of decentralization at the suburban hospital than at the urban hospital. Therefore, taking these “transferred out” individuals out of the “success” category, even though transferring out is often regarded as a positive outcome, resulted in fewer individuals at the suburban hospital with a positive outcome (and likely successful), thus conflating the urban hospital’s success with the non-success in the suburban hospital in the estimated odds ratio for hospital success. The decentralization of TB services, and DR-TB care specifically in South Africa, has been integrated into the national guidelines, to provide more patient-centered care closer to patients’ homes and to be more cost-effective while showing similar rates of treatment success [25,26].

Although more than half of the cohort reported no income at all, this is consistent with the South African demographic survey, which reports that 42% of the population in the Eastern Cape is in the country’s lowest wealth quintile and that 45% of all households receive a social grant [27]. It is interesting that the household contact status was significantly different among the three groups—more individuals who had untreated HIV lived alone. This is consistent with the HIV literature reporting that treatment support is critical for PLWH, and perhaps even more important when someone is co-infected with DR-TB. A recent meta-analysis across eight countries found that patients with DR-TB who received social support—specifically material support combined with other social support interventions (informational, emotional, and companionship)—had improvements in treatment success [28]. Finally, although we did not measure nutrition beyond a crude baseline BMI measure, nutritional support is also a well-established predictor for treatment success [29]. Understanding which modifiable factors associated with unsuccessful outcomes could be intervened upon is critical to help tailor future interventions, especially for individuals co-infected with HIV and DR-TB and any factors at the systems level.

Adverse events are common among patients on DR-TB treatment, and even more likely when there is polypharmacy, such as with ART. One study found that although adverse events are common, they were no more frequent or more severe among those co-treated for MDR-TB and HIV, and given the favorable treatment outcomes, such as those in this study, ART should not be delayed in patients with MDR/HIV co-infection [30]. However, another meta-analysis of 37 studies found that HIV infection increased the risk of adverse events in patients with DR-TB by 12% [31]. The increased risk of adverse events was primarily due to ART use rather than HIV-related immunosuppression, and the researchers recommended increased pharmacovigilance with routine monitoring, especially for patients co-infected with HIV, to ensure the timely identification and treatment of adverse events. Future studies should include longitudinal monitoring of patients to provide more detail regarding treatment outcomes, such as the severity of both DR-TB and HIV disease, and the time-varying variables, such as viral load, CD4 count, adverse events, and changes in regimen composition.

## 5. Limitations

As with all studies, there were some limitations. First, many of the variables were self-reported and could have been misclassified or some information could have been withheld, specifically with regard to more sensitive variables, such as tobacco and alcohol use. Second, because the HIV and ART statuses were the main predictors of interest, we were unable to adjust for HIV status in the regression analyses; as discussed above, the significance of treatment success depending on the site could have been conflated due to the small group sizes and issues in point estimation. Our cohort was reduced in size as we were missing 21 patient outcomes (still on treatment, missing, or moved out); the robust documentation of outcomes can be difficult, and yet ensuring the accuracy of outcome data in routine clinical practice is important when conducting retrospective research. An additional limitation is that this study did not assess the adverse events associated with DR-TB and ART therapy, which are common among individuals on DR-TB treatment and warrant further investigation in future studies. Finally, another limitation of this study is the lack of detailed data. For example, the treatment regimen only included “short course” and “long course” rather than the exact composition of drug combinations; however, the regimen compositions in South Africa are quite standard at treatment initiation. Additionally, considerable information regarding the stage of HIV/AIDS of patients—CD4 count and viral load data—was only available in a small subset of PLWH, and therefore, could not be assessed in this cohort. Future studies should prioritize gathering more detailed information and conducting longitudinal studies, in order to capture time-varying variables. We also could not include pharmacologic data on potential reasons for unsuccessful treatment outcomes. Prior meta-analyses have reported better treatment outcomes associated with the use of linezolid and later-generation fluroquinolones, bedaquiline, clofazimine, and carbapenems for DR-TB [32]. This is an area for future research, as new and repurposed medicines are being rolled out.

## 6. Conclusions

Antiretroviral therapy is imperative for patients with DR-TB and HIV co-infection to have successful treatment outcomes. Although the importance and life-saving nature of ART have been known for decades, few studies have investigated specifically the effect of HIV and ART statuses on DR-TB outcomes in a large cohort across two hospitals in a high-burden HIV and DR-TB setting. Preventing, diagnosing, and treating HIV infection could all support DR-TB treatment success. Additionally, hospitals could assess the reasons for unfavorable outcomes among their patients and work on the modifiable factors to improve facility-level care.

## Figures and Tables

**Figure 1 viruses-15-02242-f001:**
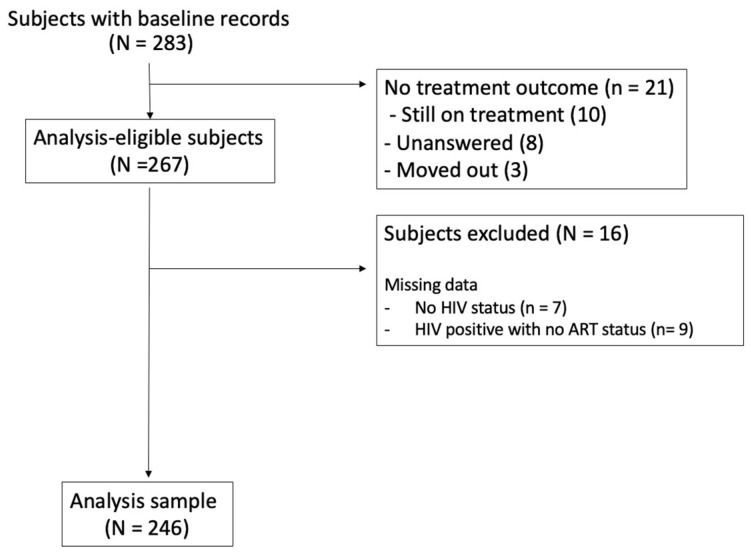
Population flow chart.

**Table 1 viruses-15-02242-t001:** Demographics of patients with DR-TB at two Eastern Cape hospitals according to their HIV and ART statuses (N = 246).

	Total (N = 246)	HIV− Patients (N = 89)	HIV+ on ART (N = 137)	HIV+ Not on ART (N = 20)	*p*-Value
Age, mean (SD)	36.0 (13.4)	35.2 (17.6)	36.1 (10.1)	38.5 (11.4)	0.43
Age					<0.01
0–15	9 (3.7%)	8 (9.0%)	1 (0.7%)	0 (0.0%)	
16–35	123 (50.0%)	44 (49.4%)	70 (51.1%)	9 (45.0%)	
36–50	81 (32.9%)	19 (21.4%)	55 (40.2%)	7 (35.0%)	
>50	33 (13.4%)	18 (20.2%)	11 (8.0%)	4 (20.0%)	
Sex					0.12
Male	134 (54.5%)	56 (62.9%)	69 (50.4%)	9 (45.0%)	
Female	112 (45.5%)	33 (37.1%)	68 (49.6%)	11 (55.0%)	
Type of DR-TB					0.13
Rif resistant	95 (38.6%)	38 (42.7%)	45 (32.9%)	12 (60.0%)	
Mono or poly resistant	2 (0.8%)	0 (0%)	1 (0.7%)	1 (5.0%)	
MDR (confirmed and not confirmed)	79 (28.5%)	25 (28.1%)	41 (29.9%)	4 (20.0%)	
Pre-XDR	22 (8.9%))	11 (12.4%)	11 (8.0%)	0 (0.0%)	
XDR	55 (22.4%)	15 (16.9%)	37 (27.0%)	3 (15.0%)	
Unanswered	2 (0.8%)	0 (0%)	2 (1.5%)	0 (0.0%)	
Previous TB drug history					0.11
New patient (no previous history)	115 (46.8%)	49 (55.1%)	54 (39.4%)	12 (60.0%)	
Previous treatment with 1st-line drugs	105 (42.7%)	31 (34.8%)	67 (48.9%)	7 (35.0%)	
Previous treatment with 2nd-line drugs	1 (5.0%)	8 (9.0%)	9 (6.6%)	1 (5.0%)	
Unanswered	8 (3.3%)	1 (1.1%)	7 (5.1%)	0 (0.0%)	
Classification of disease					0.20
Pulmonary	198 (80.5%)	73 (82.0%)	105 (76.6%)	20 (100.0%)	
Extrapulmonary	1 (0.4%)	0 (0%)	1 (0.7%)	0 (0.0%)	
HTN					0.09
Yes	32 (13.0%)	18 (20.2%)	12 (8.8%)	2 (10.0%)	
No	210 (85.4%)	70 (78.7%)	123 (89.8%)	17 (85.0%)	
Diabetes					0.01
Yes	18 (7.3%)	12 (13.5%)	4 (2.9%)	2 (10.0%)	
No	228 (92.7%)	77 (86.5%)	133 (97.1%)	18 (90.0%)	
Hearing loss					0.49
Yes	18 (7.3%)	8 (9.0%)	10 (7.3%)	0 (0.0%)	
No	172 (69.9%)	61 (68.5%)	98 (71.5%)	13 (65.0%)	
Of those who are HIV-positive, on cotrimoxazole					<0.01
Yes	118 (75.6%)	0 (0%)	108 (79.4%)	10 (50.0%)	
No	24 (15.4%)	0 (0%)	16 (11.8%)	8 (40.0%)	
Baseline BMI (mean, STD)	19.6 (4.5)	19.5 (4.0)	19.5 (4.8)	20.0 (4.6)	0.58
Pregnant					0.26
Yes	2 (1.8%)	0 (0%)	1 (1.5%)	1 (9.1%)	
No	88 (79.3%)	28 (87.5%)	52 (76.5%)	8 (72.7%)	
GeneXpert completed					0.04
Yes	117 (47.6%)	54 (60.7%)	55 (40.2%)	8 (40.0%)	
No	55 (22.4%)	16 (18.0%)	33 (24.1%)	6 (30.0%)	
TB microscopy completed					<0.01
Yes	169 (68.7%)	62 (69.7%)	98 (71.5%)	9 (45.0%)	
No	34 (13.8%)	19 (21.4%)	12 (8.8%)	3 (15.0%)	
First-line LPA completed					0.091
Yes	95 (38.6%)	40 (44.9%)	51 (37.2%)	4 (20.0%)	
No	79 (32.1%)	31 (34.8%)	41 (29.9%)	7 (35.0%)	
Unanswered	72 (29.3%)	18 (20.2%)	45 (32.9%)	9 (45.0%)	
Second-line LPA completed					0.15
Yes	88 (35.8%)	34 (38.2%)	50 (36.5%)	4 (20.0%)	
No	86 (35.0%)	36 (40.5%)	43 (31.4%)	7 (35.0%)	
Unanswered	72 (29.3%)	19 (21.4%)	44 (32.1%)	9 (45.0%)	
TB culture					<0.01
Yes	151 (61.4%)	56 (62.9%)	89 (65.0%)	6 (30.0%)	
No	48 (19.5%)	23 (25.8%)	20 (14.6%)	5 (25.0%)	
First-line drug sensitivity test (DST)					0.04
Yes	54 (22.0%)	23 (25.8%)	29 (21.2%)	2 (10.0%)	
No	134 (54.5%)	53 (59.6%)	72 (52.6%)	9 (45.0%)	
Unanswered	58 (23.6%)	13 (14.6%	36 (26.3%)	9 (45.0%)	
Second-line drug sensitivity test (DST)					0.05
Yes	56 (22.8%)	20 (22.5%)	33 (24.1%)	3 (15.0%)	
No	131 (53.3%)	55 (61.8%)	68 (49.6%)	8 (40.0%)	
Unanswered	59 (24.0%)	14 (15.7%)	36 (26.3%)	9 (45.0%)	
Regimen type					
Short regimen *	96 (39%)	45 (18.29%)	46 (18.7%)	5 (2.0%)	0.04
Long regimen *	118 (48.16%)	32 (13.06%)	74 (30.20%)	12 (4.90%)	0.03
Hospital					<0.01
Urban	141 (57.3%)	64 (71.9%)	77 (56.2%)	0 (0%)	
Suburban	105 (42.7%)	25 (28.1%)	60 (43.8%)	20 (100.0%)	
Education					0.03
No school	15 (6.1%)	6 (6.7%)	5 (3.7%)	4 (20.0%)	
Some school	210 (85.4%)	74 (83.2%)	123 (89.8%)	13 (65.0%)	
Unanswered	21 (8.5%)	9 (10.1%)	9 (6.6%)	3 (15.0%)	
Mean number years of school (STD)	8.9 (3.1)	8.6 (3.2)	9.2 (2.9)	7.5 (4.3)	0.25
Income					
Salary wages	26 (10.6%)	6 (6.7%)	18 (13.1%)	2 (10.0%)	0.31
Casual wages	14 (5.7%)	4 (4.5%)	8 (5.8%)	2 (10.0%)	0.66
UIF (grant)	2 (0.8%)	2 (2.3%)	0 (0.0%)	0 (0.0%)	0.17
No income	135 (54.9%)	45 (50.6%)	79 (57.7%)	11 (55.0%)	0.58
Self-employed	2 (0.8%)	1 (1.1%)	1 (0.7%)	0 (0.0%)	0.87
Number of dependents living at home (mean, STD)	1.73 (1.62)	1.78 (1.94)	1.75 (1.36)	1.2 (1.69)	0.51
Any household contacts					0.03
Yes	236 (95.9%)	87 (97.8%)	132 (96.4%)	17 (85.0%)	
No	10 (4.1%)	2 (2.3%)	5 (3.7%)	3 (15.0%)	
Smoker					0.01
Yes	86 (35.0%)	41 (46.1%)	40 (29.2%)	5 (25.0%)	
No	148 (60.2%)	44 (49.4%)	92 (67.2%)	12 (60.0%)	
Alcohol use					0.39
Non-drinker	126 (51.6%)	47 (53.4%)	67 (49.3%)	12 (60.0%)	
Light (1× month)	46 (18.9%)	14 (15.9%)	28 (20.6%)	4 (20.0%)	
Moderate (1× week)	32 (13.1%)	10 (11.4%)	19 (14.0%)	3 (15.0%)	
Heavy (daily)	14 (5.7%)	3 (3.4%)	11 (8.1%)	0 (0.0%)	

* South Africa adheres to a fairly standardized “short course” or “long course” DR-TB therapy approach, depending on the initial drug resistance profile. The short course consists of 4 to 6 months of moxifloxacin, amikacin, ethionamide, clofazimine, high-dose isoniazid, pyrazinamide, and ethambutol, followed by 5 months of moxifloxacin, clofazimine, pyrazinamide, and ethambutol. Meanwhile, the long course consists of 18 to 24 months including daily injectable aminoglycoside treatment for the first 6 months; or as of June 2018, an all-oral long course for 18–20 months was endorsed by the WHO and offered in South Africa, including levofloxacin/moxifloxacin, bedaquiline, and linezolid [19].

**Table 2 viruses-15-02242-t002:** Treatment outcomes among patients with DR-TB according to their HIV and ART statuses.

	Everyone (N = 246)	HIV− (N = 89)	HIV+ on ART (N = 137)	HIV+ Not on ART (N = 20)
Success	144 (58.5%)	64 (71.9%)	75 (54.7%)	5 (25.0%)
Cure	118	49	64	5
Completed treatment	26	15	11	0
Non-success	51 (20.7%)	10 (11.2%)	33 (24.1%)	8 (40.0%)
Failed	1	0	1	0
Died	39	6	25	8
Lost to follow-up	11	4	7	0
Transferred out	51 (20.7%)	15 (16.9%)	29 (21.2%)	7 (35.0%)
Censored	21	6	12	3
Still on treatment	10	1	8	1
Missing	8	5	3	0
Moved out	3	0	1	2

**Table 3 viruses-15-02242-t003:** Estimated odds ratios for success versus non-success of drug-resistant TB treatment outcomes.

Estimated OR for Success vs. Non-Success of DR-TB Treatment Outcomes		
Characteristic	OR (95% CI)	*p*-Value
Age (years) (continuous)	0.99 (0.97, 1.02)	0.57
0–35 (versus >35)	0.80 (0.42, 1.54)	0.51
Female sex	1.08 (0.57, 2.07)	0.82
HIV status		
HIV-negative	10.24 (2.79, 37.61)	<0.01
HIV+, on ART	3.64 (1.11, 11.95)	<0.05
HIV+, not on ART	Ref	–
Urban hospital—JPH (vs. suburban hospital—MPH)	6.97 (3.46, 14.04)	<0.001
Number of years of education (continuous)	1.02 (0.92, 1.14)	0.68
BMI (continuous)	1.05 (0.96, 1.16)	0.26
Smoker (vs. non-smoker)	0.87 (0.44, 1.71)	0.68
Alcohol use		
Non-drinker	Ref	
Drinker (mild/mod/heavy)	0.69 (0.37, 1.32)	0.26
Income		
Any income (salary, casual wages, grant, disability)	0.55 (0.23, 1.30)	0.17
No income	Ref	

**Table 4 viruses-15-02242-t004:** Estimated odds ratios for success versus death of drug-resistant TB treatment outcomes.

Estimated OR for Success vs. Death of DR-TB Treatment Outcomes		
Characteristic	OR (95% CI)	*p*-Value
HIV status		
HIV-negative	13.06 (3.21, 53.11)	0.0003
HIV+, on ART	4.10 (1.22, 13.72)	0.0223
HIV+, not on ART	Ref	–

**Table 5 viruses-15-02242-t005:** Estimated odds ratios for transfer out versus non-success of drug-resistant TB treatment outcomes.

Estimated OR for Transfer Out vs. Non-Success of DR-TB Treatment Outcomes		
Characteristic	OR (95% CI)	*p*-Value
Age (years) (continuous)	1.00 (0.98, 1.04)	0.57
0–35 (versus >35)	0.67 (0.31, 1.47)	0.32
Female sex	1.88 (0.86, 4.13)	0.11
HIV status		
HIV-negative	1.71 (0.47, 6.24)	0.41
HIV+, on ART	1.00 (0.32, 3.11)	0.99
HIV+, not on ART	Ref	–
Urban hospital (versus suburban hospital)	0.39 (0.16, 0.99)	0.047
Number of years of education (continuous)	1.04 (0.91, 1.20)	0.56
BMI (continuous)	1.09 (1.00, 1.19)	0.061
Smoker (vs. non-smoker)	0.47 (0.19, 1.14)	0.093
Alcohol use		
Non-drinker	Ref	
Drinker (mild/mod/heavy)	0.79 (0.36, 1.72)	0.55
Income		
Any income (salary, casual wages, grant, disability)	0.93 (0.34, 2.51)	0.88
No income	Ref	

## Data Availability

The data presented in this study are available upon request from the corresponding author.

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
