# Peer review of "The Effect of HIV and Antiretroviral Therapy on Drug-Resistant Tuberculosis Treatment Outcomes in Eastern Cape, South Africa: A Cohort Study"

_viruses, 2023, doi:10.3390/v15112242_

Round 1

Reviewer 1 Report

Comments and Suggestions for Authors

The presented article is of some interest, since the treatment of tuberculosis associated with HIV infection is a problem not only in African countries. A significant number of such patients are located in Eastern Europe and some Asian countries. Therefore, the relevance of the article is very high. The experience of managing and treating such patients should be replicated and disseminated. The methods used in the work are adequate to the tasks set in the study. Table 1 is very difficult to understand, but it may not be easy to present these results in a different way. Of course, all patients co-infected with HIV-TB should receive retroviral therapy in addition to antibiotic therapy. The article is written in good English. The reviewer has no serious comments on the text and structure of the article. Despite the fact that the conclusions of the article are quite obvious, it can be published without significant changes.

Author Response

Thank you for these comments. We appreciate your critical review of our manuscript.

Reviewer 2 Report

Comments and Suggestions for Authors

This retrospective chart review was designed to analyze the effect of HIV infection and antiretroviral therapy (ART) on DR-TB outcomes. The authors conclude that antiretroviral therapy is essential for patients with DR-TB and HIV co-infection to achieve a successful treatment outcome. Simultaneous treatment of HIV and TB is possible, but is associated with an increased risk of side effects, and some drugs are incompatible. The frequency of adverse effects of ART may be correlated with the percentage of patients who are transferred out of the study. The authors' conclusions could be strengthened by including a comparative analysis of the number of adverse events in their study.

Author Response

Thank you for bringing up the important conversation of adverse events of ART and TB treatment. Unfortunately, our dataset did not include side effect information, so we were unable to assess the rate of adverse events among this population. However, we have included this limitation in our “Limitations” and added 2 additional references into the “Discussion” regarding adverse events. Transfer out is unlikely due to adverse side effects – typically transfer out is a positive outcome when a patient is very stable and doing well and is able to transfer down to district management, rather than specialized DR-TB hospital level care; if anything, patients with adverse side effects are more likely to have stayed in this study longer as they would have been more complex patients requiring closer monitoring from this higher level of care.

We added to the Limitations: “An additional limitation is that this study did not assess adverse events associated with DR-TB and ART therapy, which are common among individuals on DR-TB treatment and warrants further investigation in future studies.”

We also added to the Discussion: “Adverse events are common among patients on DR-TB treatment, and even more likely when there is polypharmacy, such as with ART. One study found that, although adverse events are common, they were no more frequent nor more severe among those co-treated for MDR-TB and HIV, and given favorable treatment outcomes, like this study, ART should not be delayed in patients with MDR/HIV coinfection. However, another meta-analysis of 37 studies found that HIV infection increased the risk of adverse events in patients with DR-TB by 12%. The increased risk in adverse events was primarily due to ART use rather than HIV-related immunosuppression and recommends increased pharmacovigilance with routine monitoring, especially for patients coinfected with HIV to ensure timely identification and treatment for adverse events. Future studies should include longitudinal monitoring of patients to provide more granularity regarding treatment outcomes such as severity of both DR-TB and HIV disease and the time-varying variables such as viral load, CD4 count, adverse events, and changes to regimen composition.

Reviewer 3 Report

Comments and Suggestions for Authors

1. The unsuccessful treatment result mainly depends on death. Very rare failure and almost no difference of lost to follow up. The analysis only using death or not in comparison to success may have been more informative. 

2. Treatment regimen needs to be discussed when treatment result of TB is the dependent variable, especially how far the treatment regimen is standardized across the hospitals (I am afraid that suburban hospitals might not follow the standard regimen or suburban hospital doctors might give up potent drugs when mild adverse reaction occurs).

3. Considering that death is the main factor of unsuccessful treatment, HIV seriousness (viral load, CD4 count, other opportunistic infection) is necessary.

Small points

Tables 1 ; from left to right is HIV+without ART, HIV+with ART, HIV- and table 2; from left to right is HIV-, HIV+with ART, HIV+ without ART. The order had better to be the same for the readers to understand easily.

Table 2; HIV+ with ART N=37 is mistaken.  

Author Response

Response 3.1 Thank you. We have added a sensitivity analysis (Table 3a) comparing success versus only death to assess differences across the three groups.  

Response 3.2 

Thank you for this observation. Although the hospitals were urban and suburban; they are both at the same level of care – centralized, specialized drug-resistant TB hospitals. Unfortunately, this was a cross-sectional study, so we do not have information on if, or when, a regimen was changed; we only report the initial treatment regimen at initiation. South Africa adheres to quite standardized “short course” or “long course” DR-TB therapy depending on initial drug-resistance profile. The short course consists of: 4 to 6 months of moxifloxacin, amikacin, ethionamide, clofazimine, high-dose isoniazid, pyrazinamide, and ethambutol, followed by 5 months of moxifloxacin, clofazimine, pyrazinamide, and ethambutol. Meanwhile, the long course consists of: 18 to 24 months including daily injectable aminoglycoside treatment for the first 6 months; or as of June 2018, an all-oral long-course for 18-20 months was endorsed by the WHO and offered by South Africa, including levofloxacin/moxifloxacin, bedaquiline and linezolid. However, our study period crosses this 2018 timeline, and our reporting mechanisms were unable to identify at the granular level, which individual medications were prescribed nor for how long. Therefore, previously we did not feel confident in our ability to comment on treatment regimen at length, and thus, left it out of this paper as we felt it went beyond the scope of our primary analysis (citations: https://www.ncbi.nlm.nih.gov/pmc/articles/PMC7656447/).  We have added “Treatment regimen” into Table 1 to show among the three groups the rate of short-course and long-course regimens and hope this is satisfactory. We also added this to the limitations and an area for additional future work to be considered.

We have also added to the Limitations: “Another limitation of this study is the lack of data granularity. For example, treatment regimen only included “short course” and “long course” rather than the exact composition of drug combinations; however, regimen composition in South Africa is quite standard at treatment initiation. Additionally, considerable information regarding the stage of HIV/AIDS of patients – CD4 count and viral load data were - only available in a small subset of PLWH, and therefore, unable to be assessed in this cohort. Future studies should prioritize gathering more granular information and longitudinal studies, in order to capture time-varying variables.

Response 3.3 Thank you for this astute observation. We agree viral load, CD4, and other opportunistic infections (beyond DR-TB) would enhance this study. Unfortunately, these were not available in our dataset. We have added this to our “Limitations” as mentioned in the above statement and briefly in the Discussion as well.  

Response 3.4 Thank you for this comment. We have changed Table 1 to be similar to Table 2 with both Tables reading HIV-, HIV+with ART, and HIV+ without ART.

Response 3.5 Thank you for catching this typo. We have fixed this and changed it to N=137.

Round 2

Reviewer 3 Report

Comments and Suggestions for Authors

I think it can be published in the present form.